# Coastal Erosion and a Characterization of the Morphological Dynamics of Arauco Gulf Beaches under Dominant Wave Conditions

Mauricio Villagrán [1,*], Matías Gómez [1] and Carolina Martínez [2,3,4]

1   Department of Civil Engineering, Universidad Católica de la Santísima Concepción, Alonso de Ribera 2850, San Andrés, Concepción 4090541, Chile
2   Instituto de Geografía, Facultad de Historia, Geografía y Ciencia Política, Pontificia Universidad Católica de Chile, Santiago 7820436, Chile
3   Centro de Investigación Para la Gestión Integrada de Desastres Naturales (CIGIDEN), Proyecto 1522A0005 Fondap 2022, Santiago 7820436, Chile
4   ANID—Millennium Science Initiative Program—Instituto Milenio en Socio-Ecología Costera (SECOS), ICN2019_015, Santiago 8331150, Chile
*   Correspondence: mvillagran@ucsc.cl

**Abstract:** Sandy coastlines in Chile currently have strong erosive tendencies. However, little is known about the morphodynamics of these coastlines; such knowledge would allow us to understand coastline changes and incorporate this knowledge into coastal management. Accordingly, the historical scale of coastal erosion and the morphodynamic characteristics of six beaches of the Arauco Gulf, central-southern Chile (36° S), were analyzed to determine the prevailing wave conditions during winter and summer. Historical changes in the relative position of the coastline were determined using DSAS v5.1. The coupled WAVE-FLOW-MOR modules of the Delft3D 4.02 software package were used for the morphodynamic analysis. Using image processing, it was established that erosion predominates in winter seasons for almost every beach analyzed. However, the Escuadrón beach presents this trend both in winter and summer, with rates of up to −0.90 m/year (2010–2021). In addition, accretion was observed in both stations at Tubul beach. On the other hand, numerical models for the dominant conditions predict accretion in the beaches of Escuadrón, Chivilingo, and Arauco, stable conditions for Coronel beach, and erosion in Llico.

**Keywords:** coastal erosion; shoreline; littoral processes; coastal management

## 1. Introduction

Coasts are some of the most critical dynamic features of the earth's surface [1–3]. Marine-coastal areas are considered highly productive ecological resources, and pave the way for socio-economic activities around the world [2,4]. However, they are also one of the fastest-changing areas due to natural and anthropogenic stressors. Among the latter category, urbanization and urban growth stand out; by 2050, 70% of the world's population will live in urban areas [5,6]. The increased population and the location of dysfunctional economic activities on the coast have altered the natural dynamics of these areas; these changes are expressed in terms of habitat loss, reductions in biodiversity, pollution, and, most particularly, coastal erosion [7]. Among these varied consequences, coastal erosion attracts the most attention; it is a pronounced problem around the world, due to the increased frequency of coastal storms that are linked to climate variability and global environmental change, such as typhoons, hurricanes, tropical and extratropical cyclones, and coastal storms [8–11]. In this context, it becomes a priority to have available predictive models that guide adaptation processes in the face of adverse climate change scenarios, and which regulate activities that act as anthropic stressors under an integrated coastal management approach.

As such, given the lack of knowledge available about the hydrodynamic and morphodynamic processes of the coast and the coastal ocean, as well as the spatio-temporal changes in the coastline, it is of high importance to improve environmental management and coastal development [1,12–14]. For this reason, in the last few decades, significant progress has been made in studies exploring coastal processes through observation, physical experiments, and numerical models [13]. Some of these methodologies include empirical equations to determine the equilibrium profile [15], and the use of incident waves affected by diffraction to determine the coastline in beaches or bays [16,17], as well as the analysis of the coastline displacement through the use of satellite images [18]. Additionally, empirical equations have been used to determine sediment transport [19–21], and, more recently, numerical models have been used (e.g., Delft3D, xBeach, SMC, Mike21, Coupled Ocean-Atmosphere-Waves-Sediment Transport COAWST, among others).

However, due to the complexity of coastal morphological processes, sediment transport processes resulting from currents and waves are neither fully understood nor adequately described by mathematical analysis. Nonetheless, the use of quasi-3D morphological models with vertically averaged 2D model features and wave, tidal, and current effects could be a feasible tool for long-term and large-scale morphodynamic simulations [13,22–24]. Furthermore, Bertin et al. [3] and Gómez and Aránguiz [25] argue that morphodynamic models hold great potential, but they are underused in attempts to understand the physical processes that drive morphological change. Moreover, the implementation of sound coastal zone management strategies requires reliable information on erosion and/or deposition processes [26], and community and ensemble open-source models are becoming the norm to predict uncertainties [27].

In Chile, coastal erosion is not widely documented [11]. However, studies have recently established a critical erosive trend for 80% of the country's sandy coastlines, linked to the more significant recurrence and intensity of coastal storms that, since 2015, have generated extensive damage to the coast [11,28,29]. Furthermore, many beaches whose width increased due to coastal uprisings caused by the earthquake of 27/F, 2010, are registering erosive processes and sudden morphological changes [11,30,31]. Regarding these changes, Gomez et al. [32] characterized the coastal sediment transport direction on seven beaches in the Arauco Gulf using the SWAN model under predominant wave and storm conditions. However, their study had a qualitative approach because they did not quantify the transported sediment. Meanwhile, Martinez et al. [33] analyzed the impact on the morphodynamics of the Tubul beach of the uplift of the terrain during the earthquake of 27 February 2010 (Mw = 8.8) through numerical modeling. Their results show a change in the morphological response of the beach for regular wave conditions, increasing the tendency to accretion on the post-tsunami beach.

The present study analyzes coastal erosion for historical scales, and the morphodynamics of the main Arauco Gulf beaches (see Figure 1) for dominant wave conditions. A combination of characteristic waves and wind in winter and summer conditions and tides is included in the analysis, which was conducted using Delft3D quasi-3D software. Moreover, this study aims to address the current gap in the available literature regarding the morphodynamic evolution of the Arauco Gulf by understanding the main sediment transport trends in the study area, as well as the processes of sedimentation and seasonal erosion. As such, this study will forge scientific bases for integrated coast management, generated in consideration of current climate change scenarios.

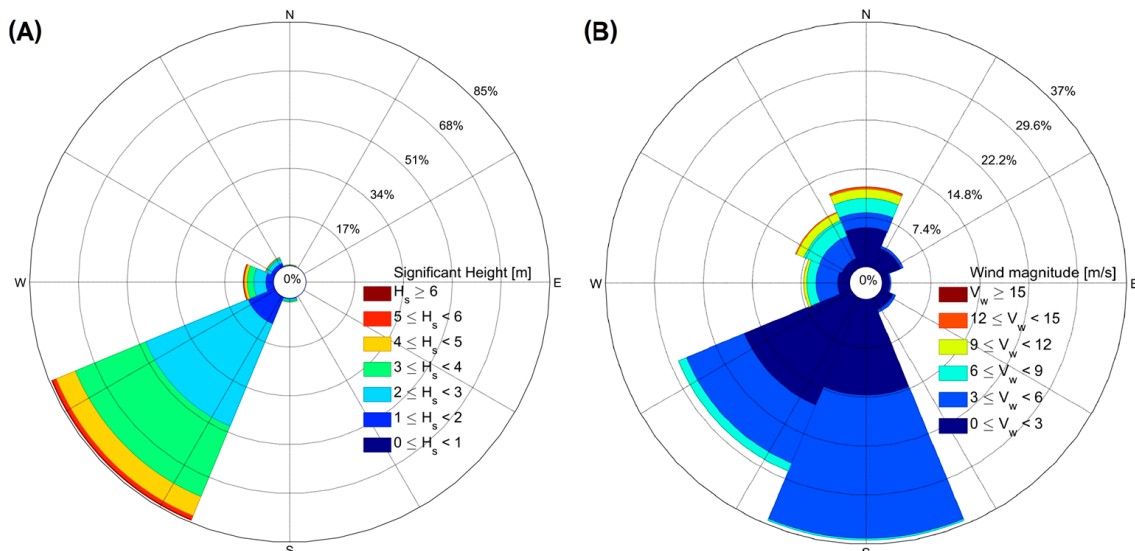

**Figure 1.** Wave and wind rose for: (**A**) WaveWatchIII (37° S 74° W node), and (**B**) the Arauco Gulf.

## 2. Study Area

### 2.1. Arauco Gulf Morphology

The coastal area of the Arauco Gulf (see Figure 2) is a coastal plain located in south-central Chile (37.10° S, 73.3° W). This geomorphological structure, a product of block faults, is characterized by a sandy plain in the upper part of the Mesozoic–Cenozoic unit [34]. The Arauco Gulf comprises a group of beaches, including Escuadrón, Coronel, Chivilingo, Arauco, Tubul, and Llico (from north to south). The Holocene sands in the coastal plain have different origins and compositions. From the Bíobio River (in front of the submarine canyon) to Coronel, the sediment is black, of Andean origin. Furthermore, it is characterized by having a fine to medium size with clasts of basaltic composition due to its origin, as the product of the volcanic activity associated with the Antuco volcano in the valley of the Laja River during the late Pleistocene and Holocene [34]. Meanwhile, from the south of Coronel, the sands are thin with a coastal origin and a light color, deriving mainly from small rivers and coastal basins, which erode intrusive granite and quartzite typically found in the sector [34].

The Arauco Gulf has a clear north–south orientation. It is bounded to the east by the mainland and, to the west, by Santa Maria Island. Santa Maria Island has an area of 35 km²; it is 11.5 km long in a north–south direction, and its width ranges between 0.5 and 6.5 km in an east–west direction.

Moreover, to the north of the Arauco Gulf, the Bíobio canyon can be found. It is described as a submarine valley of steep walls, whose head begins at a depth between 15 and 20 m. The head of the canyon is located further south of the current mouth of the Bíobio River, at a distance of approximately 300 m in the area closest to the coast (Escuadrón Beach) [34]. The canyon extends to the northwest, up to the oceanic trench, where it reaches a depth of 4500 m. At the edge of the head of the canyon there is a significant contribution of sediments (mainly black sands) from the Bíobio River, which are deposited progressively, especially during periods of river flooding. It is confirmed that the canyon is in a phase of erosion, which produces a migration of its headwaters towards the mainland, endangering the sediment supply at Escuadrón beach [34].

### 2.2. Wave Climate

The wave climate in the Arauco Gulf is dominated by waves generated on the west wind belt (40–60° S), which propagate through the Pacific Ocean [35]. The intense surface winds associated with extratropical cyclones are the main generators of incident waves on the Arauco Gulf. The wave regime in the Arauco Gulf is influenced by the South

Pacific anticyclone, which generates prevailing S–SW winds during the year. In winter, the northward shift of the Pacific anticyclone enhances the growth of low-pressure systems, triggering strong winter storms [11]. The prominent waves in the Arauco Gulf typically show a significant wave height of 2.9 m, a peak period of 12.8 s, and a peak direction of 232.4°. Meanwhile, the dominant wind magnitude and wind direction are 4.71 m/s and 228.7°, respectively.

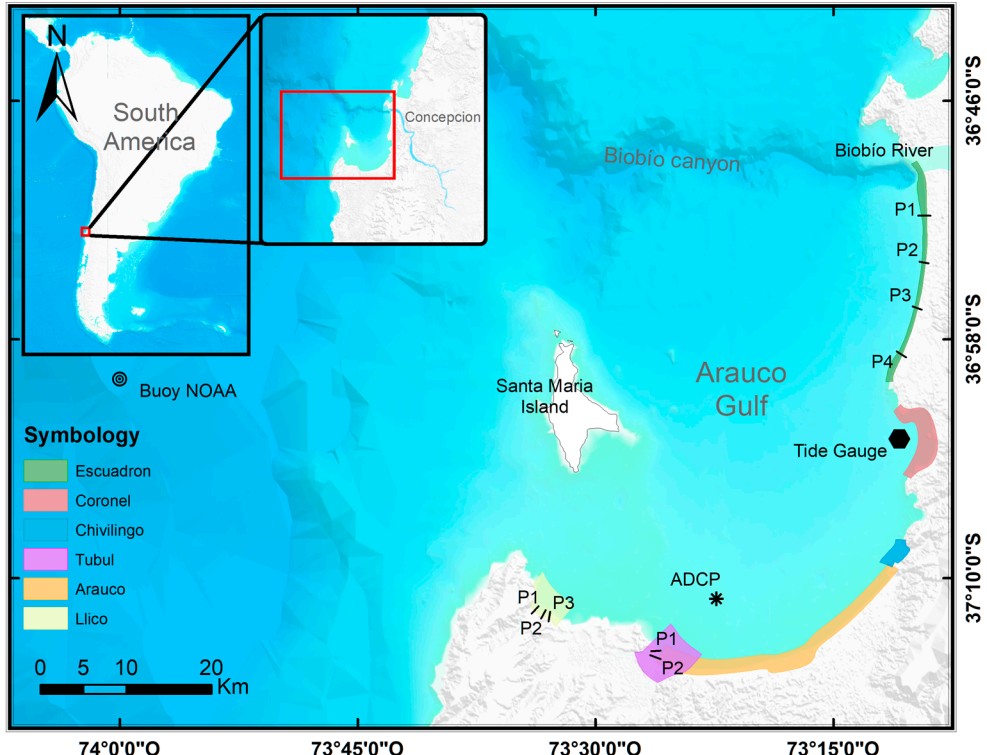

**Figure 2.** Location of the studied beaches along the Arauco Gulf. Cross-shore sections are illustrated in solid black line, NOAA virtual buoy with circles, colonel's tide gauge with hexagon and ADCP with asterisk. P1–P4 are control transects at every beach.

## 3. Methods

### 3.1. Coastal Erosion Rates

Changes in the shoreline position of five beaches were determined from topographic surveys, aerial photographs, satellite images, and survey maps using the U.S. Geological Survey's (USGS) Digital Shoreline Analysis System (DSAS) [36], in combination with ArcGIS 10. First, the annual rates of change in shoreline positions were determined along several transects within each beach using the Linear Regression Rate method (LRR). Then, mean rates of change for each beach were computed as the average of the rates obtained from each profile. The LRR calculates the rate of change by fitting a least squares regression line to all the points in the shoreline for each transect. The method is considered easy to use and apropriate for the number of beaches and transects under review [37,38].

The maximum high tide visible in aerial photographs/satellite images or recorded in topographic surveys was used as a proxy in identifying the shoreline. This limit generally coincides with the beginning of the foredune, where the rear beach is stabilized. Available historical aerial photographs of these beaches were georeferenced and complemented with detailed topographical surveys and Google Earth Pro satellite images. The error in the shoreline position from the aerial photographs and satellite images was estimated based on the criteria of Pixel Representativeness and Mean Square Error/RMSE, thus ensuring errors less than 1 m (Table 1).

**Table 1.** Aerial photographs used.

| Beach | Flight | Source | Year | Month | Scale |
|---|---|---|---|---|---|
| Escuadrón | FONDEF 20 | SAF | 1992 | 2 | 30,000 |
| Escuadrón | FONDEF 20 | SAF | 1998 | 2 | 15,000 |
| Arauco—Laraquete | FONDEF 20 | SAF | 1992 | 5 | 20,000 |
| Arauco—Laraquete | - | Survey | 2010 | 6 | 20,000 |
| Coronel | - | Google Earth | 2017 | 9 | 30,000 |
| Coronel | - | Google Earth | 2013 | 9 | 30,000 |
| Coronel | - | SAF | 1992 | 11 | 20,000 |
| Coronel | - | SAF | 1982 | 11 | 30,000 |
| Coronel | - | Google Earth | 2021 | 8 | 2500 |
| Coronel | - | Google Earth | 2021 | 2 | 2500 |
| Coronel | - | Google Earth | 2018 | 1 | 2500 |
| Coronel | - | Google Earth | 2014 | 1 | 2500 |
| Llico | FONDEF 20 | SAF | 1992 | 5 | 30,000 |
| Llico | - | Survey | 2010 | 7 | 20,000 |
| Tubul | Chile 30 | SAF | 1983 | 1 | 30,000 |

Finally, the rates of change were classified according to the four categories of coastal evolution trend proposed by Rangel et al. [38]: high erosion (<−1.5 m/year), erosion (between −0.2 and −1.5 m/year), stability (between −0.2 and +0.2 m/year) and accretion (>+0.2 m/year).

*3.2. Wave Climate*

Deep-water wave data (37° S 74° W node) were acquired every 3 h using the Wave-WatchIII numerical model, as there is no deep-water wave monitoring system available in the region. In addition, the NOAA virtual buoy (see Figure 2) was analyzed between 1997 and 2018, characterizing the waves according to energy content in the winter and summer seasons.

*3.3. Numerical Model*

For the development of this study, the WAVE-FLOW-MOR modules of the Delft3D 4.02 software [39] were coupled with a time frame of 30 h, with the first 6 h being used as a warm-up. Moreover, the bathymetry of the sector was constructed from the GEBCO database, nautical charts, and detailed bathymetry obtained in the DIN 13-2011-UCSC project [40].

Delft3D consists of several process modules that simulate wave propagation, currents, sediment transport, morphological changes, and water quality in coasts, rivers, and estuaries. The hydrodynamic and dynamic sediment processes resulting from waves were simulated with the Delft3D model, coupling the WAVE [41] and FLOW [42] modules in a quasi-stationary mode (see Figure 3). The process mentioned previously involves the bidirectional coupling of a non-stationary hydrodynamic calculation in combination with standing wave simulations. In addition, WAVE was activated every 10 min of the hydrodynamic simulation. Afterwards, it performed a stationary simulation using the water levels, currents, and bed levels provided, and calculated using the hydrodynamic model as boundary conditions. The WAVE module, in turn, feeds the hydrodynamic model with the radiation stress.

3.3.1. WAVE Module Setup

The WAVE module is based on the third-generation SWAN model [43,44]. Deep water physics was simulated using the combination of wind input, white capping [45], and quadruplet nonlinear interactions [46]. In the case of shallow waters, nonlinear triad interactions [47] and bottom friction using the JONSWAP scheme [48], both set by default in SWAN, were considered. On the other hand, the model proposed by Battjes and Janssen [49] was used to simulate dissipation due to depth-induced wave breaking. Finally, diffraction

was considered in the wave calculation, based on the phase-decoupled refraction-diffraction approximation [50].

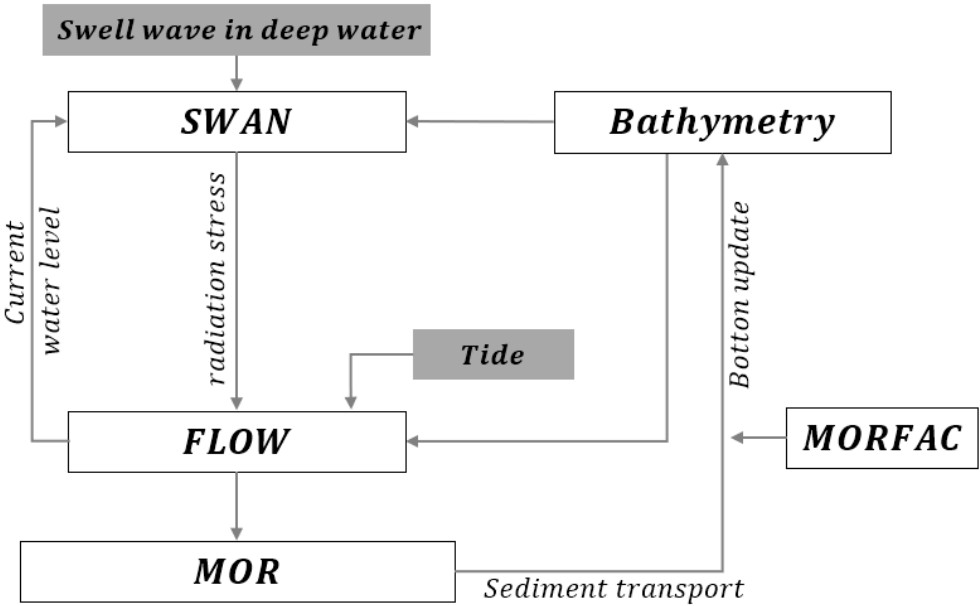

**Figure 3.** Online updating scheme of the Delft3D modules.

Deep-water boundary conditions were obtained from the wave characterization (see Table 2), while the FLOW module provided the (depth-averaged) current velocity, the sea level, and bed bottom variations.

**Table 2.** Mean erosion rates for beaches in the Arauco Gulf across different time periods.

| Winter Station | Period | N° Years | Average Erosion (m/year) | Category |
|---|---|---|---|---|
| Escuadrón | 2016–2021 | 5 | 0.01 | stable state |
| Escuadrón | 1978–2021 | 43 | −0.37 | erosion |
| Escuadrón | 2010–2021 | 11 | −0.58 | erosion |
| Coronel | 2013–2021 | 8 | −0.42 | erosion |
| Arauco—Laraquete | 1992–2021 | 29 | −0.85 | erosion |
| Arauco—Laraquete | 2010–2021 | 11 | −0.22 | erosion |
| Tubul | 2011–2021 | 10 | 0.57 | accretion |
| Llico | 2010–2021 | 11 | −2.35 | high erosion |
| **Sumer Station** | | | | |
| Escuadrón | 1992–2017 | 25 | −0.32 | erosion |
| Escuadrón | 1955–2022 | 67 | −0.09 | stable state |
| Escuadrón | 2010–2021 | 11 | −0.90 | erosion |
| Coronel | 1982–2017 | 35 | −0.02 | stable state |
| Arauco—Laraquete | 1992–2017 | 25 | 1.22 | accretion |
| Arauco—Laraquete | 2016–2022 | 6 | 1.22 | accretion |
| Tubul | 1983–2017 | 34 | 0.56 | accretion |
| Tubul | 1961–2022 | 61 | 0.34 | accretion |
| Tubul | 2010–2022 | 12 | 1.66 | accretion |
| Llico | 2012–2022 | 10 | 0.34 | accretion |
| Llico | 1992–2017 | 25 | 0.82 | accretion |

### 3.3.2. WAVE Grids

Curvilinear grids of the WAVE model were nested in order to capture tidal phenomena, wave propagation, and wind-driven flow, and their interactions. Each beach model consists of two nested, well-structured, orthogonal curvilinear grids for the Gulf Ocean and beaches.

The grids are aligned to the land edges to improve the numerical representation of the grid versus flow solution, and they extend into deep water (11 km west of Santa Maria Island). The upper WAVE grid has a resolution of 150 × 120 m and is used for all beaches, while the detail grids for each of the beaches have a resolution of 50 × 40 m (see Figure 4).

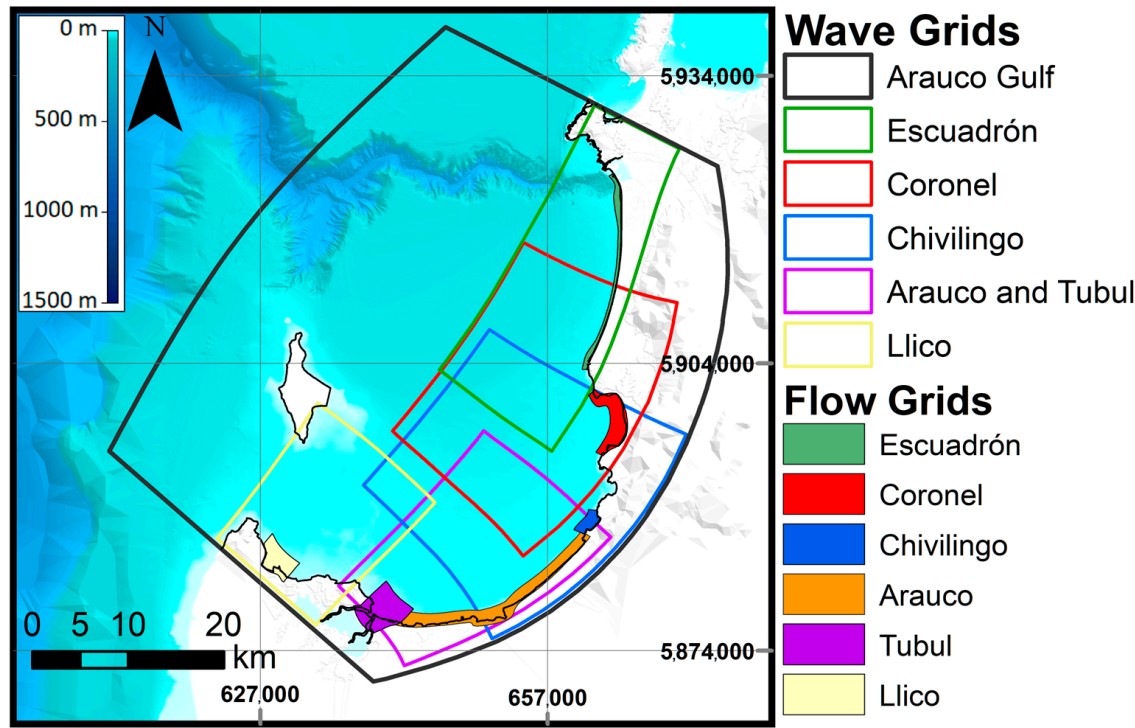

**Figure 4.** Grid configuration: non shaded contours represent nested grids for each of the beaches in WAVE, while shade contours correspond to detailed grids of the beaches in FLOW.

### 3.3.3. FLOW Module Setup

The FLOW module solves the equations of motion in two and three dimensions. It is based on a system of continuity equations, horizontal momentum, transport for conservative constituents, and the turbulence closure model. In addition, the vertical momentum equation reduces the hydrodynamic pressure relationship, since it assumes that the vertical accelerations are small compared to the gravitational acceleration. The aforementioned factors make the Delft3D model suitable for predicting flow in shallow seas, coastal areas, estuaries, lagoons, rivers, and lakes [42].

In order to preserve numerical stability in the model, the time step of each FLOW domain was adjusted according to a maximum courant number of 0.6. The hydrodynamics were generated by the radiation tensor delivered by the wave module [51], the wind present in the domain, and sea level variations. The radiation tensor is proposed by the WAVE module every 10 min. The wind condition is fixed and is obtained from the wave characterization. Furthermore, the sea level variations are imposed as an astronomical boundary condition on the seaward side for each beach model (along the coast), comprising 15 constituents. In contrast, for the edges perpendicular to the coastline, the Neumann boundary condition [52] was used to allow the homogeneous input of the wave corresponding to the tide. Constituents were determined through harmonic sea level analysis of the Coronel tide gauge (http://www.ioc-sealevelmonitoring.org/station.php?code=crnl2, accessed on 14 October 2017) over a two-month time frame (June to July 2018) using the tide tool.

Conversely, sediment transport and morphological variation were determined using the morphodynamic model in the FLOW module. A characteristic sediment diameter of

D50 = 0.853 mm was used [53]. In addition, to consider the combined effects of waves and currents on sediment transport, the Van Rijn [54] equation was selected.

Finally, to simulate the changes in the bottom of the bed as a result of sediment transport, a MORFAC morphological factor of 30 was used. This decision was taken because of the 31 h time frame being used, and because, within that time frame, the first 7 h are essentially a warm-up period. Additionally, the effect of the change of the bed because of sediment transport as a consequence of hydrodynamics is amplified by 30, having the monthly variations in the bed's bottom. Thus, the hydrodynamic and morphodynamic analyses consider 1 and 30 days of simulation, respectively. Using the MORFAC factor improves computational time, allowing for a reduction in a large percentage of the simulation time [55]. However, following the MORFAC approach could change the order of events, and possible conflicts may arise in combination with limited sediment availability and bed stratigraphy simulations [42]. In order to use MORFAC correctly, it was ensured that the bed variation was less than 5% in order for it not to affect the accuracy of the morphodynamic model, as suggested in [42].

### 3.3.4. FLOW Grids

Flow grids correspond to the curvilinear grids, with a domain limited at the land edge by the contour line 4 m above mean sea level and at the sea edge up to the 20 m isobath (see Figure 4). In addition, they have a resolution of 20 × 20 m and are refined on the axis perpendicular to the coastline between the 5 m isobath and 2 m above mean sea level.

### *3.4. Model Calibration*
### 3.4.1. WAVE Calibration

In order to perform the calibration of the WAVE module, spectral swell data for the interior of the Arauco Gulf were used. These wave data were measured with an Acoustic Doppler Current Profiler ADCP (brand Nortek, Model Signature 250, Carlsbad, CA, USA, see Figure 2) between 22 September 2017, and 6 October 2017, recording the prevailing waves and storm waves. Furthermore, to analyze the model behavior, the root mean square error (RMSE) and correlation coefficient were used for the significant height and peak period. Meanwhile, the CircStat tool v1.21 [56] was used for the peak direction.

The wind data were also extracted from the WaveWatchIII model (37° S 74° W virtual node). Due to the morphology of this area, the wind magnitude was modified to attenuate the peak winds during the storms and those winds coming from directions in which the Gulf is sheltered.

### 3.4.2. FLOW Calibration

The calibration of the FLOW-MOR module consisted of comparing the simulated and observed transverse profiles of the Escuadrón, Tubul, and Llico beaches between two successive profile measurements, using as forcing factors the average swell or storm conditions between the dates of the selected profiles. For Escuadrón, the months of March and April 2011 were selected. In the case of Tubul, the profiles measured between March and May 2012 were used, and for Llico, the periods selected were February 2013 and February 2014.

The evolution of the transverse profile of the beach depends on the bottom and suspended transport generated by currents and waves, which is calculated using the Van Rijn [54] equation. Therefore, the resulting profile was adjusted by varying the suspended and bottom sediment transport factors resulting from the effect of currents and waves belonging to the Van Rijn [54] equation. The variation in these factors ranged between 0.1 and 0.6, because [57] suggests that values outside this range give unrealistic profiles.

## 4. Results

### 4.1. Coastal Erosion

The changes in the coastline position determined through the DSAS are presented below. The results indicate that the beaches of the Arauco Gulf show a general tendency to erosion during the winter, across different time periods; this tendency is more critical in Escuadrón and Llico (Table 2). During the summer, most beaches show accretion, considering different time scales, except for Escuadrón, where erosion tends to maintain a steady rate. When considering a short time scale (the last ten years), the beaches show erosion during the winter, except in Tubul; meanwhile, in summer, they exhibit accretion, except in Escuadrón.

Considering the historical trends, Escuadrón beach stands out as having the highest erosion rates. Furthermore, this state appears in all the analyzed periods, including winters and summers. Coronel Bay also tends to exhibit this same condition. The Arauco—Laraquete sandy littoral tends to erode in winter for long- and short-time scales, but in summer, this trend is reversed with higher accretion rates. Of all the beaches included in the analysis, Tubul presents the highest accretion rates in summer, considering the different time scales, and this state is maintained in winter in the short term. Llico has the highest erosion rate for short time scales, showing a reversal in the last ten years −2.35 m/year, with a much lower accretion in summer (0.34 m/year).

From a spatial point of view, the seasonal distribution of erosion rates is different on each beach. Escuadrón, Arauco—Laraquete, and Llico seem to have a more homogeneous distribution during the winter along the entire sandy coastline, while Tubul and Coronel present intercalations of areas with stability and accretion in short segments, occurring the other way around in summer (see Figures 5 and 6).

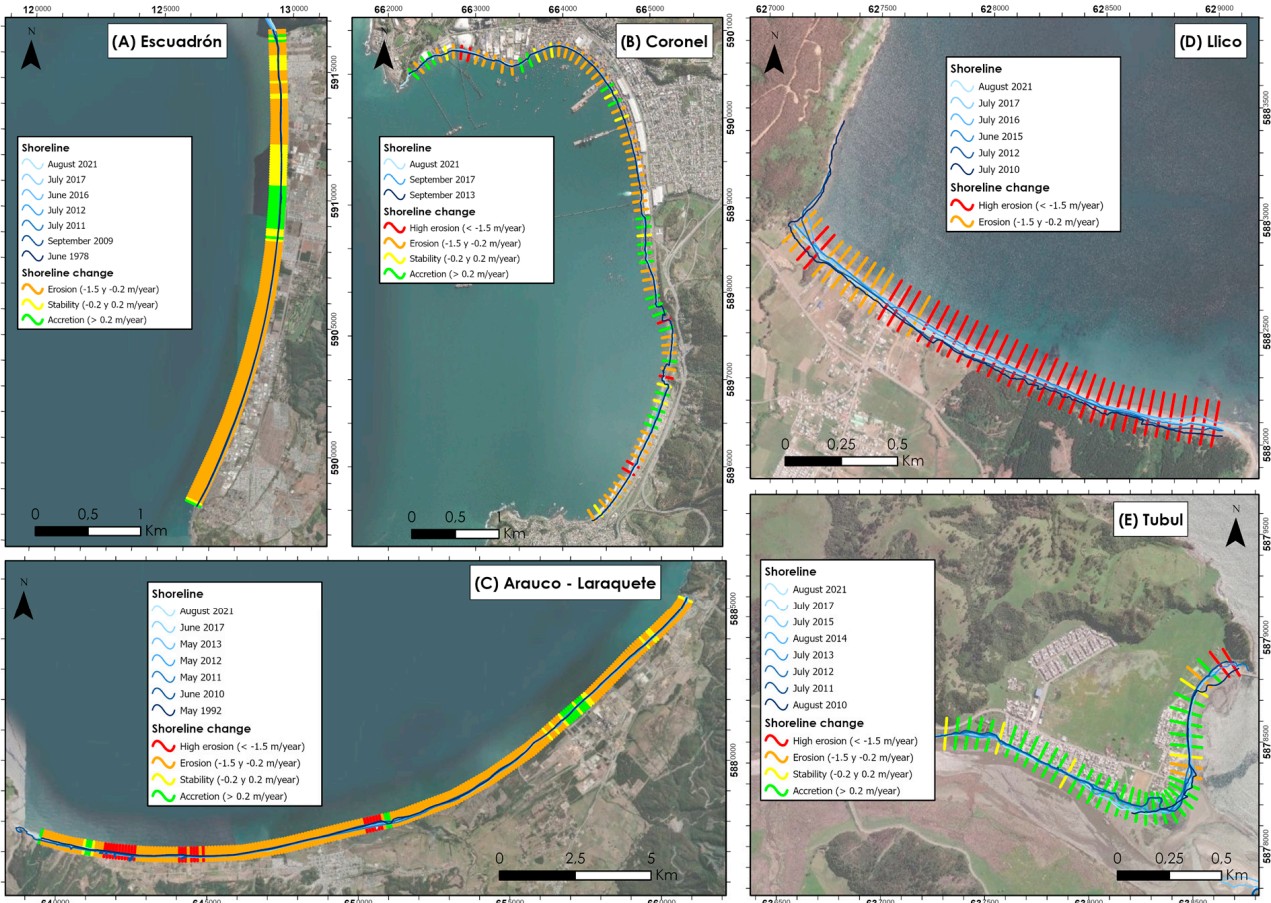

**Figure 5.** Spatial distribution of erosion rates on the Arauco Gulf beaches during winter. (**A**) Escuadrón beach, (**B**) Coronel beach, (**C**) Arauco beach, (**D**) Llico beach and (**E**) Tubul beach.

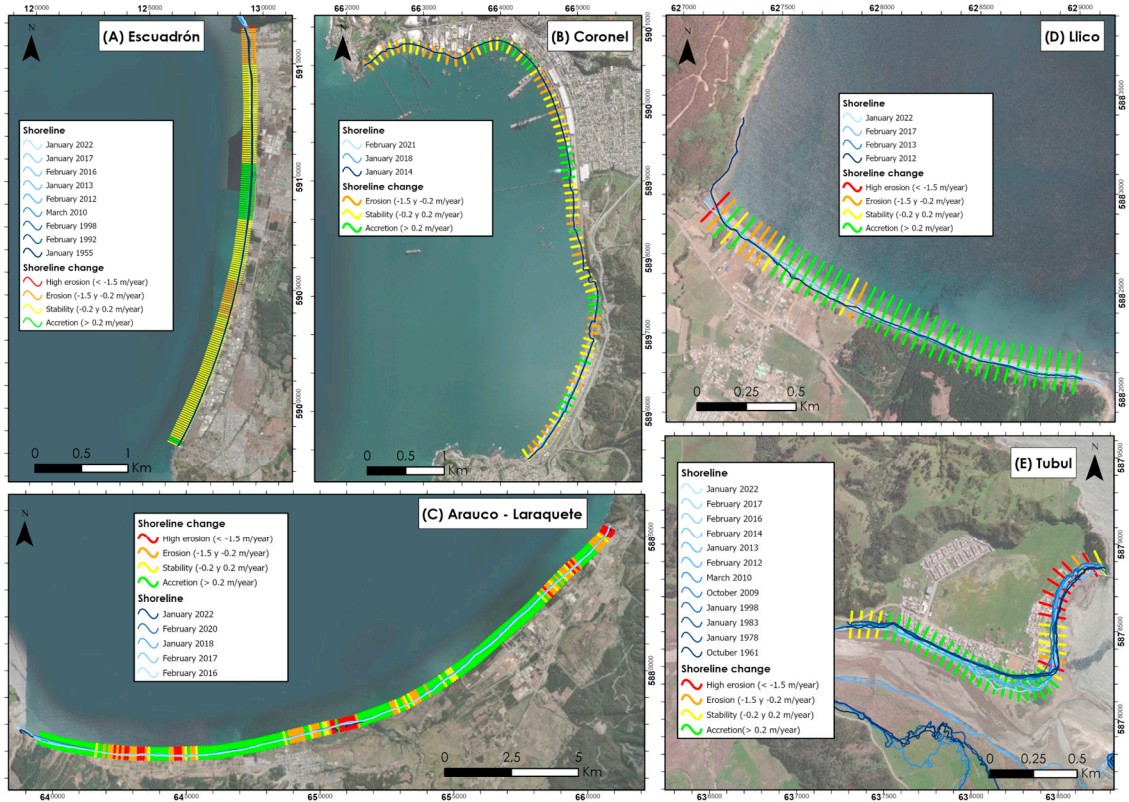

**Figure 6.** Spatial distribution of erosion rates on the Arauco Gulf beaches during summer. (**A**) Escuadrón beach, (**B**) Coronel beach, (**C**) Arauco beach, (**D**) Llico beach and (**E**) Tubul beach.

### 4.2. Wave Climate Characterization

Figure 7 shows the time series of deep-water spectral wave parameters for the 37° S 74° W node of the WWIII model for 1997–2018. Presented in gray are the total data plotted on the left axis. On the other hand, the right axis shows the averages during the winter months in the continuous blue line. Red lines show the averages during the summer months, while the segmented lines represent the data trends according to the season.

It can be observed that the incident waves in the Arauco Gulf have a high energy content, with an annual average height of 2.8 min in deep water and waves up to 8.3 m high during storms. For the summer months (October–March), the swell has a significant height, peak period, and direction of 2.6 m, 12.2 s, and 231.7°, respectively. In contrast, for the winter months (April–September), the swell conditions present a higher energy content, with a wave height of 3.0 m, a peak period of 12.6 s, and a peak direction of 233.1° (Table 3). On the other hand, there is an upward trend in significant height in summer of 0.011 m/year and, in winter, of 0.013 m/year. Furthermore, the peak period shows an increase of 0.1 s/year in both seasons, while the peak direction shows a positive increase in summer of 0.20 °/year and a negative trend in winter of −0.27 °/year.

**Table 3.** Simulation cases.

| Scenario | Hs (m) | Tp (s) | Dp (°) | Vw (m/s) | Dw (°) |
|----------|--------|--------|--------|----------|--------|
| Winter   | 3.0    | 12.6   | 233.1  | 4.9      | 242.1  |
| Summer   | 2.6    | 12.2   | 231.7  | 4.9      | 242.1  |

The 15 main constituents and tidal phases obtained from the harmonic analysis of the Coronel tide gauge are presented in Table 4. The main lunar semi-diurnal (M2) component has an amplitude of 0.45 m, a main solar (S2) component of 0.12 m, and a major elliptical lunar (N2) component of 0.11 m. On the other hand, the luni-solar (K1) and main lunar

(O1) diurnal components have amplitudes of 0.20 and 0.10 m, respectively. Finally, the monthly lunar component (MM2) has an amplitude of 0.12 m.

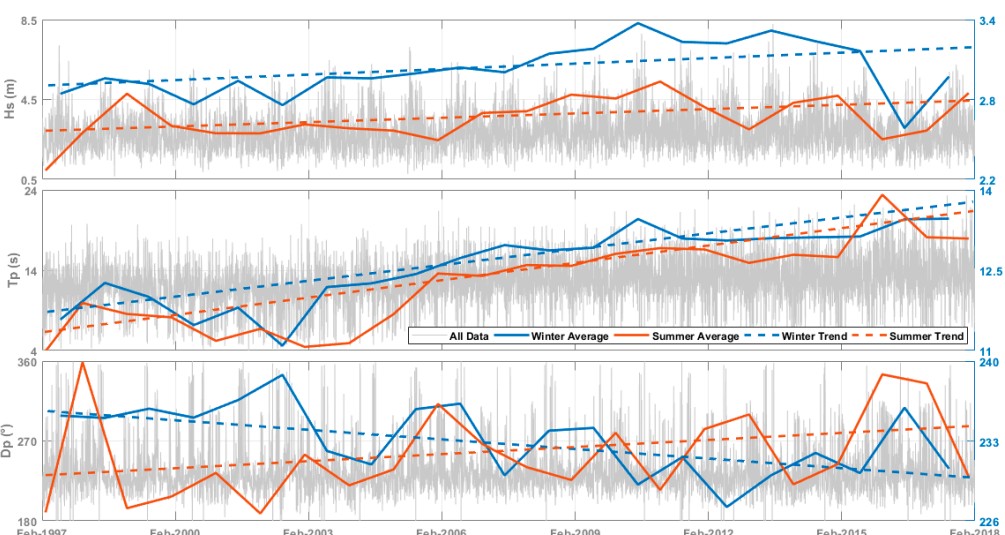

**Figure 7.** NOAA buoy spectral wave data time series (37° S 74° W) between February 1997 and February 2018.

**Table 4.** Tidal constituent derived from Coronel's tidal gauge.

| Tide (Const.) | Amp (m) | Phase (°) | Tide (Const.) | Amp (m) | Phase (°) |
|:---:|:---:|:---:|:---:|:---:|:---:|
| M2 | 0.45 | 75.7 | Q1 | 0.02 | 335.9 |
| K1 | 0.21 | 50.5 | L2 | 0.01 | 56.4 |
| S2 | 0.12 | 101.0 | J1 | 0.01 | 55.6 |
| MM | 0.12 | 265.5 | OO1 | 0.01 | 123.9 |
| N2 | 0.12 | 51.3 | ETA2 | 0.01 | 136.6 |
| O1 | 0.11 | 1.9 | NO1 | 0.01 | 19.9 |
| MSF | 0.04 | 144.2 | EPS2 | 0.01 | 351.4 |
| MU2 | 0.03 | 14.00 | | | |

Wave propagation for characteristic conditions in winter and summer is presented in Figure 8. For both scenarios, the waves come mainly from the southwest and are diffracted by Santa María Island. Therefore, although the waves have a significantly higher height outside the Gulf, the incident waves inside the Gulf have a similar energy content.

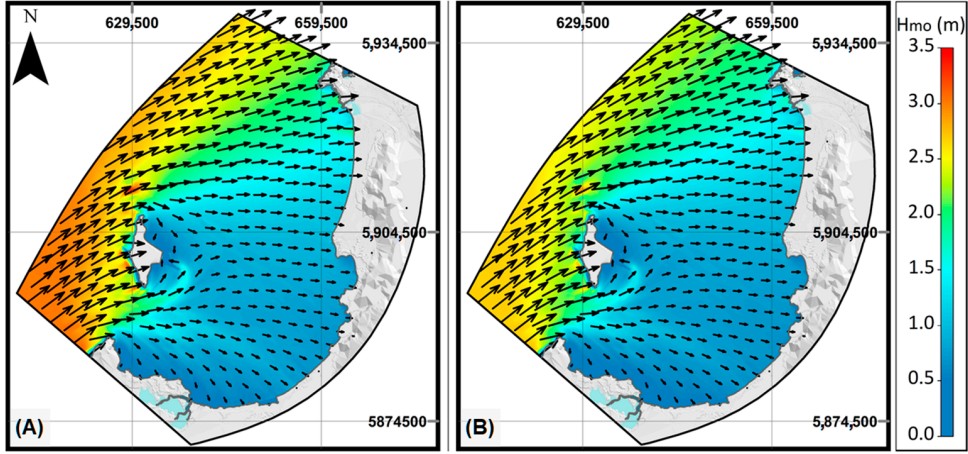

**Figure 8.** Predominant wave propagation in the Arauco Gulf: (**A**) winter, (**B**) summer. The dominant wave characteristics in deep water for summer and winter scenarios are shown in Table 3. Arrows illustrate the peak wave direction.

### 4.3. WAVE Calibration

In the first instance, the wave propagation without wind consideration (see the green line in Figure 9) presented a poor fit, especially during storms. This can be explained by dominant swell energy attenuation due to the protection provided by Santa Maria Island and the fetch for waves coming from the northwest direction in the Gulf. Then, the wind delivered by the WWIII model at the deep water node (37° S 74° W) was incorporated, obtaining an overestimation of the wave height values for normal swell conditions (see the gray line in Figure 9), as observed between September 22 and 27; meanwhile, between September 28 and October 6, the storm peaks were overestimated. In order to address this issue, a site-specific empirical adjustment was implemented to achieve the wave calibration inside the Arauco Gulf. Accordingly, the wind speed was adjusted using two factors. The first factor attenuates the winds associated with the storms, while the second factor decreases the intensity of the winds coming from the directions where the Gulf is protected. This equation is presented below:

$$V_{wind} = V - \frac{2.35V^3\overline{V}}{(2V_{max})^3} - F_0 \cdot \left(\frac{247.5}{F_1}\right)^5 \left(0.8 + \frac{V}{2V_{max}}\right)^3, \tag{1}$$

where $V$ corresponds to the wind speed obtained from the WWIII model (node 37° S 74° W) at instant t, and $\overline{V}$ and $V_{max}$ are the median and maximum wind speeds of the historical record data. In addition, the Fo factor discretizes those directions where the Gulf presents shelter, and the factor F1 attenuates winds coming between directions (350–180°).

$$F_0 = \begin{cases} 1, \ D_v < 247.5° \ v \ D_v > 350° \\ \\ 0, \ 247.5° \leq D_v \leq 350° \end{cases} \qquad F_1 = \begin{cases} 180, \ D_v < 180° \\ 220, \ 220° < D_v \leq 260° \\ 260, \ 260° < D_v \leq 350° \end{cases}$$

When using the modified wind values in the wave propagation, we obtained a plotted time series represented by a dashed orange line in Figure 8. Furthermore, the model achieves good representation under regular wave and storm conditions. Additionally, the time series for wave height Hs, peak period Tp, and peak direction Thp (see Figure 10) at the ADCP location (see Figure 2) for the simulated and observed data is illustrated.

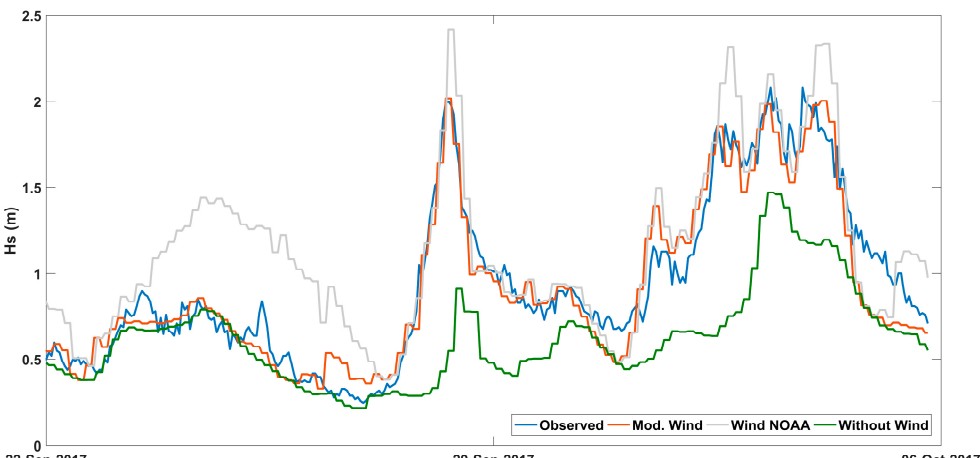

**Figure 9.** Wave height at ADCP location. The blue line shows the observed data, the green dashed line corresponds to results that do not consider the wind, the grey dashed line represents the data calculated considering the wind, as given by the WWIII model, and the orange dashed line shows the calculated data using the proposed modified wind data.

Additionally, it can be observed that the peak period manages to adjust for normal conditions (see Figure 10). However, during the storm, it shows differences due to the high variability during the event. Conversely, the simulated peak direction is different from the

observed values. This is attributed to the fact that the SWAN model does not represent diffraction well.

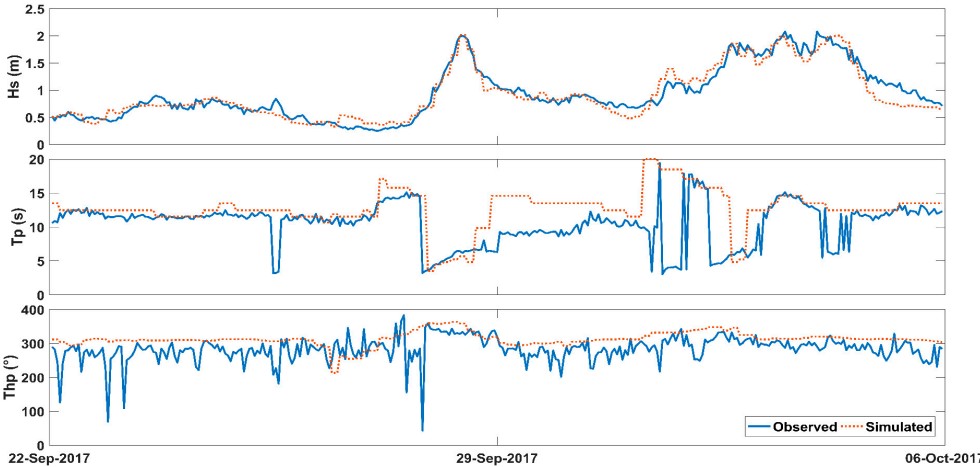

**Figure 10.** Wave data time series measured with ADCP (blue solid line) and simulated data (orange dashed line).

### 4.4. MOR Calibration

Figure 11 shows the observed and simulated transverse profiles of the Escuadrón and Tubul beaches. In Escuadrón, an accretion profile is observed under the prevailing wave conditions between March 2011 and April 2011. Meanwhile, using a storm that occurred between March 2012 and May 2012 as a forcing agent, an erosional profile is generated for Tubul beach. In the case of Escuadrón, it can be seen that the profile adjusts well in the upper zone, although in the lower zone, there are differences. In Tubul, the transverse profile observed in May 2012 shows a break due to the erosion generated by the storm, although the simulated profile does not manage to generate this discontinuity. The differences between the observed and simulated profiles arise from the model's simplification. The forcing wave data used during the simulation were constant over time, and not variable. Despite this, the resulting simulated profiles indicate that the model can represent the main erosional and accretional trends.

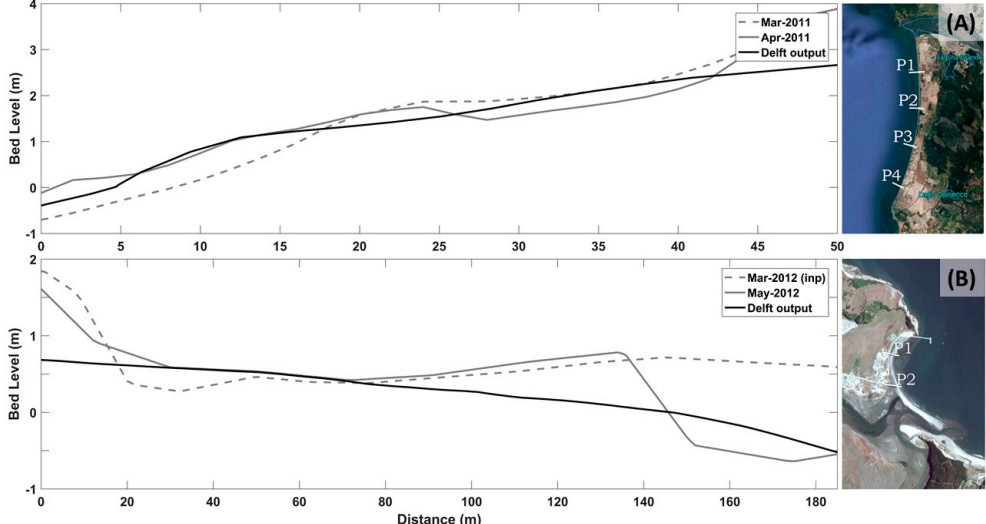

**Figure 11.** FLOW-MOR module calibration results for (**A**) the Escuadrón profile 2 with: Hs = 3.2 m, Tp = 13.3 s, Thp = 226.1°, (**B**) the Tubul profile 2 with Hs = 5.5 m, Tp = 10.8 s, Thp = 341.4°. The initial profiles and calculated profiles are shown in dashed lines and solid lines, respectively. P1–P4 illustrate the control transects at beaches of Escuadrón and Tubul, respectively.

### 4.5. Seasonal Coastal Morphodynamic

The Escuadrón beach, under predominant winter and summer climate conditions, shows accretion in the dry zone of the beach (see Figure 12). For the predominant winter climate, there is more significant variability in the bed concerning the characteristic summer climate, because the winter swell has a higher energy content. However, for both scenarios, there is a similarity in the sediment transport direction. Moreover, Escuadrón beach presents the highest rate of sediment transport in Arauco Gulf due to the contribution of sediments from the Bíobio River and the exposure of this beach to the waves.

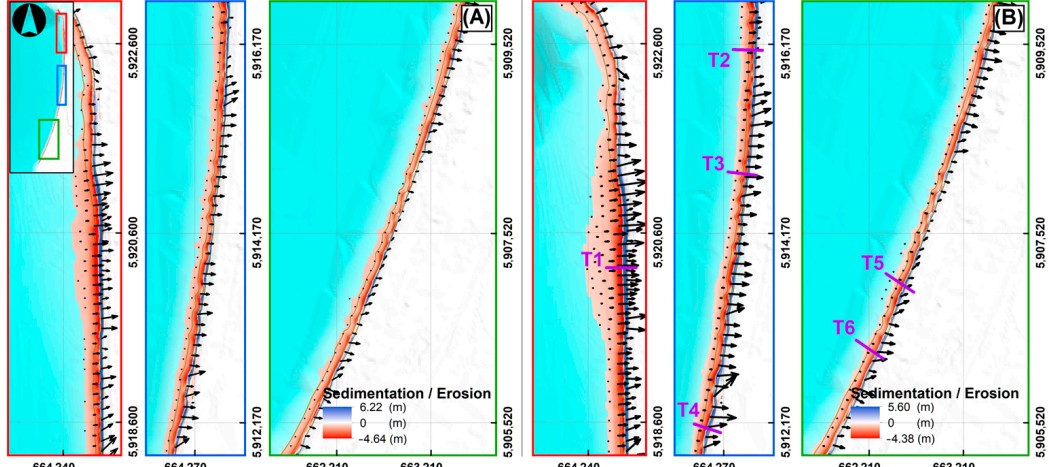

**Figure 12.** Morphological changes in Escuadrón during: (**A**) winter and (**B**) summer. Blue and red areas indicate accretion and erosion, respectively. Vectors show the average direction of sediment transport, and the gray lines represent the contours. Escuadrón beach is divided into three zones: north (red box), center (blue box), and south (green box). T1–T6 are control transects at Escuadrón beach.

Morphodynamic activity in the Coronel beach is more pronounced in the marine zone (Figure 13), and, more precisely, in the central zone of the beach. In addition, there are no significant differences in the resulting morphology between the winter and summer scenarios, reaching a maximum erosion of 2.5 m and an accretion of 3.8 m. The total transport rate of Coronel beach is generally low; thus, the beach is in a stable condition. The results indicate that the most significant morphological variations are seen in the central area of Coronel, where a rocky area can be found.

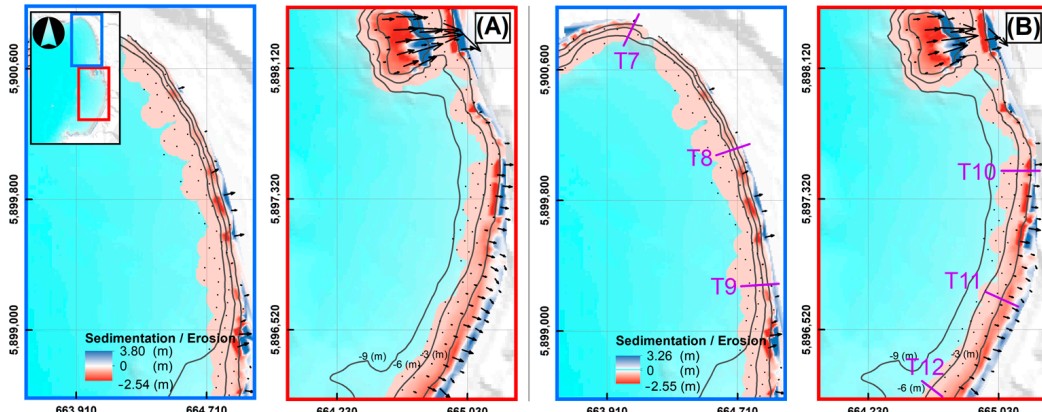

**Figure 13.** Morphological changes in Coronel during: (**A**) winter and (**B**) summer. Coronel beach is divided into two zones: north (blue box) and south (red box), as in Figure 12. T7–T12 are control transects at Coronel beach.

On Chivilingo beach, more significant morphological variability was observed in the winter scenario, reaching depths of up to five meters. On the other hand, the summer scenario only showed variations up to three meters deep (see Figure 14). In addition, the winter scenario shows a maximum erosion of 2.5 m and an accretion in the central area of the beach of 2.1 m. In comparison, the summer scenario shows erosion of 2 m near the coastline and accretion of 1.9 m. Overall, the sediment transport and morphodynamic rates of both scenarios indicate a state of beach accretion.

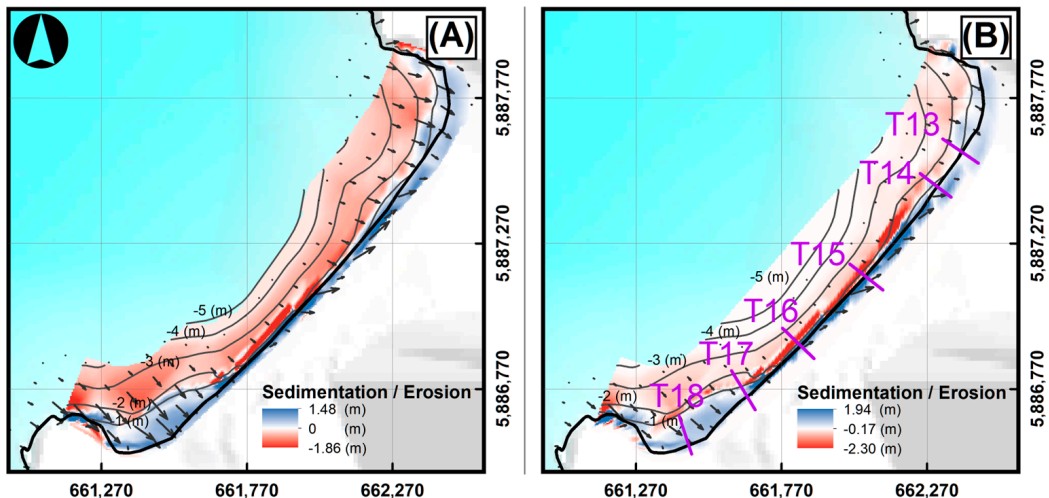

**Figure 14.** Morphological changes in Chivilingo during: (**A**) winter and (**B**) summer, as in Figure 13. T13–T18 are control transects at Chivilingo beach.

Arauco beach presents similar morphodynamic activity in both scenarios (Figure 15), reaching a maximum erosion of 2.6 m and an accretion of 1.5 m in the dry zone of the beach; therefore, Arauco beach displays accretion. In addition, there is a higher sediment transport rate in the beach's central zone (the west sector of red square, Figure 14). On the other hand, it is observed that the western zone of the beach has a higher transport rate than the northeastern zone.

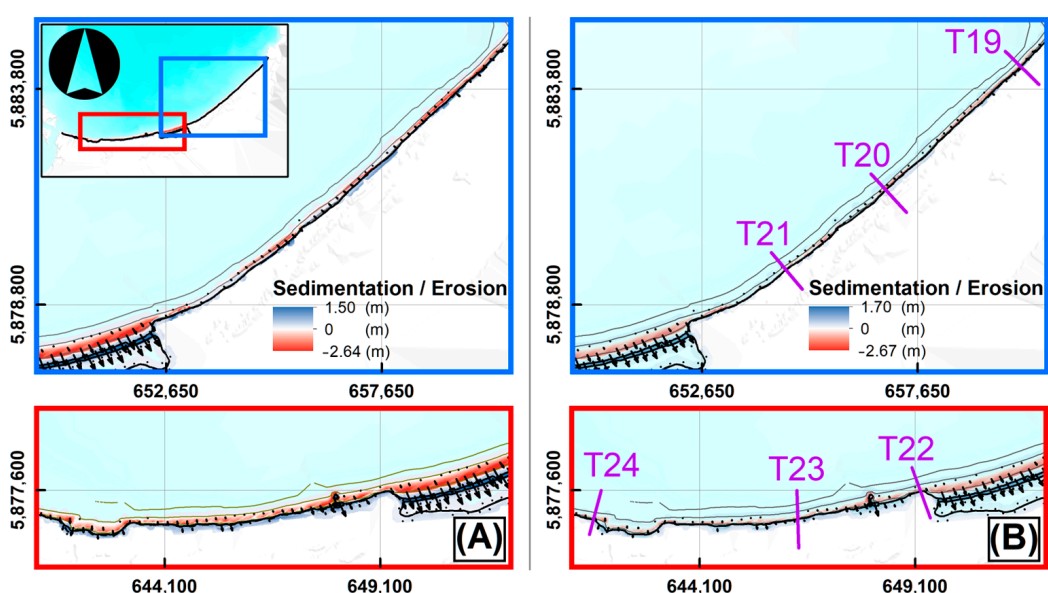

**Figure 15.** Morphological changes in Arauco during: (**A**) winter and (**B**) summer. Arauco beach is divided into two zones: east (blue box) and west (red box), as in Figure 13. T19–T24 are control transects at Arauco beach.

As seen in Figure 16, Tubul shows slight variations in morphodynamic activity between the winter and summer months, reaching a maximum erosion of 0.3 m and an accretion of 0.3 m. Meanwhile, accretion is observed along the beach and in the northwest area of the Tubul River's mouth. This indicates that the sediment is transported toward the estuary's mouth. This sediment deposition in front of the river mouth increases the sediment bar. In addition, the southeast component of the sediment transport causes the bar to move eastward.

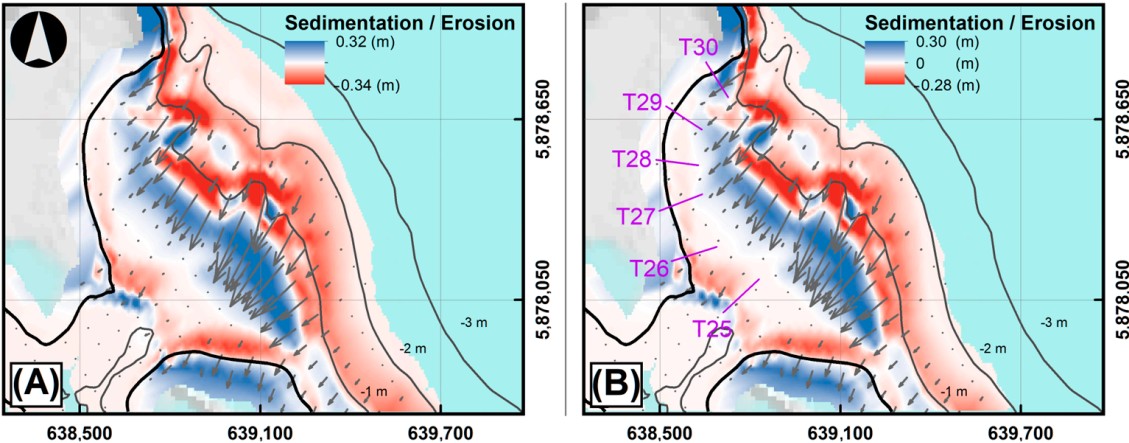

**Figure 16.** Morphological changes in Tubul during: (**A**) winter and (**B**) summer, as in Figure 13. T25–T30 are control transects at Tubul beach.

Figure 17 shows the morphological changes in Llico during winter and summer. In the northwestern area of Llico beach, an erosive profile is observed for prevailing winter swell conditions and accretion is detected for the average summer swell (Figure 17). The erosion produced in the winter season reaches a value of 0.4 m, while the accretion generated in summer corresponds to 0.3 m, meaning that Llico Beach has an erosive profile. However, there is a low sediment transport rate in the northwest area of the beach and more significant activity in the southeast of the beach.

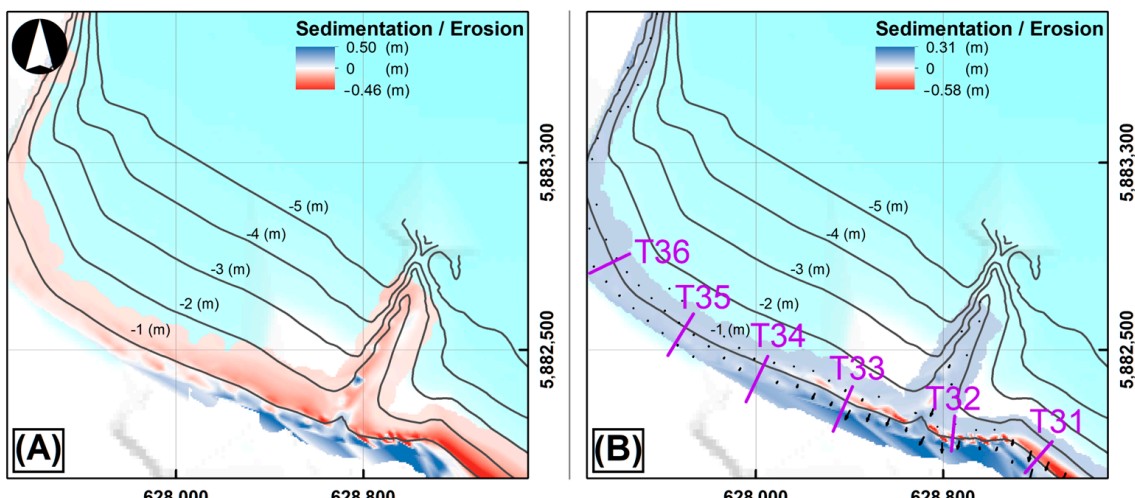

**Figure 17.** Morphological changes in Llico during: (**A**) winter and (**B**) summer, as in Figure 13. T31–T36 are control transects at Llico beach.

Figure 18 illustrate the estimated longitudinal (LST) and cross-shore sediment transport (CST) rates evaluated for all beaches in transects T1 to T36, as shown in Figures 12–17. All beaches have CST towards the dry zone (accretion), as observed in Figure 18. In Escuadrón, the LST is mainly in the northern direction. However, in the southern zone, there

is an LST in the opposite direction (south). Furthermore, the LST decreases by 25% from summer to winter (see Figure 16), and the CST decreases by 53% from summer to winter. In Coronel, the LST has a northwesterly direction during summer, but a 45% variation in a southeasterly direction can be found during winter.

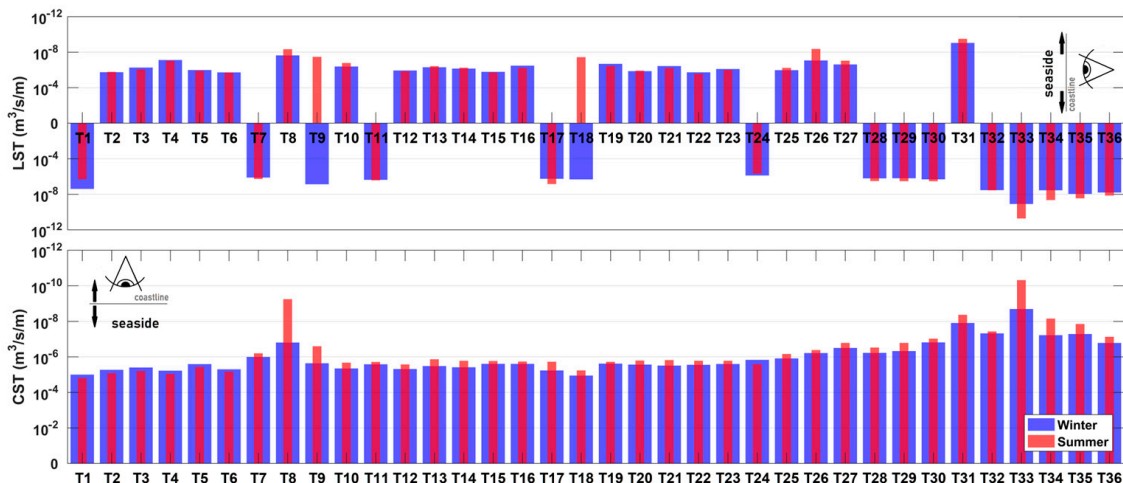

**Figure 18.** Longshore and cross-shore sediment transport. Each bar corresponds to the mean sediment transport in the profiles indicated in Figures 12–17.

Similarly, the CST increases by 49%. In Chivilingo, the LST predominates in a northwest direction during summer, although in winter, it decreases by 23%, and in the southern zone, it changes direction. Meanwhile, the CST increases by 53% from summer to winter. In Arauco, the LST predominates in the northwest direction. However, at the beach's western end, the LST has a southwest direction, and is generally higher by 56% in summer compared to winter.

On the other hand, the CST shows a decrease of 37% in winter. At Tubul, the northwestern part of the beach has a northeastward LST, while the southeastern part has a southeastward LST. Between summer and winter, the LST and CST increase by 51% and 45%, respectively. On the other hand, in Llico, the LST has a mainly northeast direction, and it is observed that the LST and CST increase by 67 and 69%, respectively, from summer to winter. Finally, it is observed in Table 3 that the average wave direction and peak period decrease by 1.4 and 0.3 s, respectively, between the winter and summer seasons, and this seasonal variation generates considerable changes in the sediment dynamics along the different beaches of the Arauco Gulf.

## 5. Discussion

The beaches located southwest of the Arauco Gulf have lower morphodynamic activity due to the shelter provided by Santa María Island, which diffracts the waves, attenuating the energy content of the incident waves on these beaches. Furthermore, Escuadrón has the most active morphodynamic activity in the Arauco Gulf. One possible explanation for this is that the energy of the incident waves is less attenuated by diffraction, together with the sedimentary contribution provided by the Bíobio River.

The most significant morphological variation in Coronel beach occurs in the central area, although this area is rocky. Unfortunately, the model does not differentiate between rocky and sandy areas, so this variability is not representative of the system. Therefore, the spatial discretization of the sediment size and the assignment of non-erodible zones in future models would allow for a better simulation of this structure.

Tubul beach presents more significant variability in the marine zone due to the 1.4 m uplift caused by the earthquake of 27 February 2010 [58], and because it has an estuarine area made up of two coastal basins (the Tubul and Raqui rivers). Furthermore, this uplift destabilized the system, so it is expected that the beach will present more significant coast-

line variability than the rest of the Arauco Gulf beaches. In light of this, Martínez et al. [33] analyzed the spatio-temporal changes in the coastline after the earthquake of 27 F/2010, establishing the steady state of the shoreline, with an average erosion rate of $-0.016$ m/year in the 1961–2017 period. However, erosion predominated in the period between these two large earthquakes (1961–2009), with an average rate of $-0.386$ m/year.

In general, the differences between the results associated with historical changes in the relative position of the coastline and trends in morphodynamic activity are due to the method used. The morphodynamic activity results only consider dominating wave conditions. In contrast, analyses of the annual changes in the coastline position consider both calm conditions and storm events, sediment extraction (Escuadrón beach), and displacements due to seismic activity in the region. Moreover, Luijendijk et al. [55] stated that morphological models based on representative wave conditions are not capable of reproducing the morphological responses of beaches, mainly because they can not capture the alongshore distribution of the longshore sediment transport. According to our results, however, they still offer a fair representation of the sediment dynamics and trends in a sector.

Some studies have shown that the region's beaches are very sensitive to the uplift and subsidence tectonics associated with earthquakes [30,33,59,60]. Moreover, the increase in the frequency and magnitude of coastal storms in Chile has generated a rise in the erosion rates of the beaches, which explains why they cannot recover [29]. Therefore, it becomes necessary to conduct a study that includes these forces, together with the anthropic activities that take place on the beaches of the Arauco Gulf, in order to improve our understanding of the beaches' dynamics, thereby improving coastal management.

## 6. Conclusions

According to the historical trends of changes in the relative coastline position calculated with DSAS on the beaches of the Arauco Gulf, for the last 30 years, erosion has occurred in Escuadrón both in winter and summer. However, on the sandy coast of Arauco—Laraquete, Coronel, and Llico, erosion occurs only in winter, while in summer, accretion predominates. Across short time scales, erosion is prevalent during winter on all beaches except Tubul.

For the analyzed period (1997–2018), the predominant swell incident in the Arauco Gulf comes from the southwest and has an average height of 2.6 m in summer and 3 m in winter. Moreover, the trends in the wave series analyzed indicate a positive trend in wave height and the peak period in both seasons. In contrast, the peak direction presents a positive trend in summer and a negative trend in winter. Finally, there is a decrease of $1.4°$ in the average wave direction between the summer and winter seasons and the peak period, which decreases by 0.3 s from winter to summer.

Wind plays a critical role in the swell propagation into the Gulf due to the fetch for waves coming from the northwest, and the attenuation of the dominant swell energy content due to the protection provided by Santa Maria Island.

For the dominating wave conditions in winter and summer, the beaches of Escuadrón, Chivilingo, Arauco, and Tubul show a state of accretion, while Coronel beach is in equilibrium. Moreover, Llico beach is in an erosive state. Thus, its community's vulnerability will increase in the face of flooding risks due to possible future extreme events. Additionally, our results indicated differences in the DSAS for the Escuadrón beaches in both seasons, and for Coronel and Arauco in winter. Based on these results, it is necessary to implement measures to mitigate erosion on the beaches of Escuadrón, Arauco, and Llico. The implementation of "hard" solutions, such as submerged breakwaters, is not feasible due to the seismic activity in the region. However, "flexible" solutions, such as sand nourishment and the installation of geotextiles, are good options to compensate for the unbalanced coastline that results from coastal erosion and anthropogenic impacts.

Although the models manage to represent the erosion and accretion phenomena on the Arauco Gulf beaches, they need to be continuously supplied with information. For

example, the contribution of the sedimentary and hydrodynamic rivers that flow into the beaches complements the calibration measurements of flow velocity in surf and sediment transport. Finally, it is necessary to generate a more detailed topobathymetry of the beaches in order to improve the representation of the hydrodynamic and morphodynamic activity of the system.

**Author Contributions:** Conceptualization, M.V.; methodology, M.V. and M.G.; data statistics, M.V. and M.G.; preparation of figures, M.G.; writing—original draft preparation, M.V. and M.G.; writing—review and editing M.V., M.G. and C.M.; funding acquisition, M.V. and C.M. All authors have read and agreed to the published version of the manuscript.

**Funding:** This research was funded by the CORFO 17CTEBI-83504, FONDECYT 1151367, FONDE-CYT 1200306, Fondap 1522A0005 grants from the Agencia Nacional de Investigación y Desarrollo (ANID) of Chile and ANID/Millennium Science Initiative Program—ICN2019_015.

**Institutional Review Board Statement:** Not applicable.

**Informed Consent Statement:** Not applicable.

**Data Availability Statement:** The data presented in this study may be available in a redacted form upon request from the corresponding author.

**Acknowledgments:** We thank the postgraduate program of the Universidad Católica de la Ssma. Concepción for its continued support. The authors would like to thank Rafael Aránguiz for providing the detailing bathymetry of the Arauco Gulf.

**Conflicts of Interest:** The authors declare no conflict of interest relevant to this study.

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
