# Peer review of "Coastal Erosion and a Characterization of the Morphological Dynamics of Arauco Gulf Beaches under Dominant Wave Conditions"

_water, doi:10.3390/w15010023_

Round 1

Reviewer 1 Report

Water

Coastal erosion and characterizing the morphological dynamics 2 of Arauco Gulf’s Beaches under dominant wave conditions

This paper presents the coastal erosion for historical scales and the morphodynamic characteristics for six beaches of the Arauco Gulf, central-southern Chile (36ºS), are analyzed. Historical changes in the relative position of the coastline were determined using DSAS v5.1. The coupled WAVE-FLOW-MOR modules of the Delft3D software were used for the morphodynamic analysis

General comments:

The plagiarism percentage is about 18%. I think it is a good percentage.

Specific comments:

1.Introduction

1.     In page 5, line (32-34:  There is a mistake in English editing and this make problem in understanding the sentences, i.e. “Among the latter, urbanization and urban growth stand out, underlining that by 2050 70% of the world's population will live in urban areas [5,6]”.  It should be” Among the latter, urbanization and urban growth stand out, underlining that by 2050, 70% of the world's population will live in urban areas [5,6]”

2.     What is the addition of this paper to the previous researches. Has any research discussed the same topic with the same methodology? Or you pose a new methodology not applied before through previous research? Or you just update the evolution of the study area? You have to specify accurately.

3.     You have to add a paragraph explaining the previous work on the study area to let the paper understand the gap that this research will fill.

2. Study area

4.     It is preferred to put a subsection for the coastal processes in the regions “  prominent wave direction, current and its direction, ….)

5.     It will be better if you add a figure illustrate the wind and wave rose.

3. Methods

1.     Regarding the acquired date of the aerial photographs, I see that it is acquired in different months. So, I think it will not give accurate results during the analysis of shoreline change rate. All images should be in the same time to ensure that the extracted shoreline will be the right one.

2.     It will be better for Figure1 if you put the coordinates of the axes outside the figure as it is interrupted with the figure details.

3.     which technique in DSAS you used and explain why?

4.     Did you consider courant number? Please add your explanation in the manuscript.

5.     Regarding the morpho dynamic modelling using delft3d, what is the duration of each simulation? . please mention it clearly in the manuscript.

4. Results

In the first paragraph, you have to mention the method you used to get the results below , this will the reader keeping follow up.

It will be better for figures if you put the coordinates of the axes outside the figures.

5- Discussion

In the second paragraph, you mentioned “Unfortunately, the model does not differentiate between rocky or sandy areas, so this variability is not representative of the system”. So how did you deal with this problem in your model?

Author Response

Dear Reviewer,

Please find enclosed the revised manuscript titled “Coastal erosion and characterizing the morphological dynamics of Arauco Gulf’s Beaches under dominant wave conditions”. The authors would like to express their sincere appreciation for the constructive comments raised by you. In addition, you can also find enclosed a detailed answers to the reviewers’ comments, as well as the corresponding actions performed to improve the original manuscript submission.

Furthermore, a proofreading process was applied in order to improve the English language of the article.

Yours Sincerely

Matías Gómez

Co-Author,

Reviewer 2 Report

The paper is interesting and generally well presented and illustrated, but the approach seems to lack sufficient insitu data to justify the numerical modelling assumptions. It is rather long, over-referenced beyond the actual scope of the study and some references are trivial/irrelevent. The english starts off very good but degenerates later, especially the overuse of casual phrases such as "on the other hand" that often don't fit the contexts.

I'm not sure that the conclusions are fully justified by the process because of the lack of insitu data. Recognising the data issues early on in the development is important to maintain the reader's respect for the numerical modelling challenges. The accuracy of the modelled results, in the context of morphological modeling generally, are greatly overstated.

Line#

25          no full stop

37          “due to the increased frequency of extreme events” – I would caution use of this casual commentary as there is no actual evidence of this globally. The use of mobile phones with cameras and 24-hr news cycles certainly “attracts attention”. More relevant for the timescale of this study would be to consider the anthroprogenic encroachment of the growth of settlements on the dynamic coastal zone (i.e. demographics).

51 - 56  This is (a) far too long and (b) not a proper sentence.

70          Refer line 37 also. Changes “since 2015” are more likely related to ENSO decadal variability, which is not addressed anywhere in the paper?

131        To have errors less than 1 m for such analyses would be practically impossible. If you had more data you would find more variability.

138        Table 1 – if these are the only times that analyses have been undertaken the confidence in results must be quite low?

141        The lack of objective offshore data is a significant drawback to the reliability the analyses – suggest mor commentary on this.

149        Reliance on the GEBCO database bathymetry seems inconsistent with the degree of numerical modelling applied to the project. Were any independent checks done of its accuracy, especially datum?

199        Only 15 constituents  from a two month sample – why so little?

204        Reference [53] is unnecessary.

217        MORFAC=30; please explain the 5% issue.

230        The phrase “In contrast” does not seem to suit the context here.

232        Similarly the phrase “Conversely” seems misplaced. Also, Ref [57] and associated discussion seems trivial and unnecessary.

237 ­- 238            This is not a sentence.

245 – 246           These are VERY short periods?

264                      Table 2, heading typo “Sumer”

295 – 296           These “rates” seem inconsequential. Refer #70, need to discuss decadal climate variability.

314                      Figure 7 – the patterns appear almost identical, especially near the beaches, with only higher energy in winter?

318                      During the “storms”? i.e. more than one? Also, the phrase “The previously mentioned …” could simply be “This”?

319                      “protection provided by Santa Maria Island and the Fetch in the Gulf”

                             (1) why is fetch capitalised?

                             (2) these are two separate issues that act in opposite ways?

320 – 332           I would recommend rewriting/removing most of this as it is a site-specific empirical adjustment that you have devised to achieve calibration that has no value to other studies. Eqn (1) and the detailed parameter selection should be replaced by a clear statement of what the changes were intended to (reasonably) replicate versus the actual site exposure.

335 – 337           This is not a sentence, and the phrase “On the other hand” does not seem appropriate.

341                      The reference to “yellow” should be “grey”?

346                      Besides “diffraction”, peak period Tp is an implicitly erratic parameter. Better to discuss a more stable parameter such as Tz. It is not clear that the ADCP site would be greatly affected by the island. Simple refraction analyses would be useful to test this.

369                      “bottom”?  Do you mean “bed” or “profile”?

373                      Suggest replace “group of beaches belonging to the” by “in”

374 – 375           Suggest replace “diffraction generated” as the mechanism by “sheltering provided”

376 – 377           I don’t understand this sentence.

383                      “Morphodynamics” is a process, so needs some qualification (is used elsewhere also). Try “Morphodynamic activity” or some such phrasing.

396                      “2.53” is rather too precise?? Likewise the other values that follow…

Figure 11 to 15                What are the vectors in these diagrams? Average wave direction and height?

428                      Typo “of”

455                      Refer comments #383

459                      Refer #374                       

476 – 493           These revelations come too late in the development and need to be foreshadowed in the introductory remarks. As it stands, they contrast unfavourably on the detail of the numerical modelling, which seems poorly supported by the lack of detailed measurements.

503                      Refer #295 – irrelevant without some ENSO considerations. Suggest remove as detracts from the overall value of the work.

508                      Refer #319 (1)

510                      Refer #374

521 – 524           Yes! This needs to be mentioned earlier when 2-digit accuracy is being suggested.

Author Response

(The authors gave the same response as above.)

Reviewer 3 Report

Major revisions are suggested. Comments and suggestions are attached in the attachment.

Author Response

(The authors gave the same response as above.)

Round 2

Reviewer 1 Report

Nothing

Reviewer 3 Report

Accept in present form.